# Coxsackievirus Protease 2A Targets Host Protease ATG4A to Impair Autophagy

**DOI:** 10.3390/v14092026

**Published:** 2022-09-13

**Authors:** Yiyun Michelle Fan, Yizhuo Lyanne Zhang, Amirhossein Bahreyni, Honglin Luo, Yasir Mohamud

**Affiliations:** 1Department of Cellular & Physiological Sciences, University of British Columbia, Vancouver, BC V6T 1Z3, Canada; 2Centre for Heart Lung Innovation, St. Paul’s Hospital, Vancouver, BC V6Z 1Y6, Canada; 3Department of Pathology and Laboratory Medicine, University of British Columbia, Vancouver, BC V6T 2B5, Canada

**Keywords:** enterovirus, autophagy, viral subversion, ATG4

## Abstract

Enteroviruses (EVs) are medically important RNA viruses that cause a broad spectrum of human illnesses for which limited therapy exists. Although EVs have been shown to usurp the cellular recycling process of autophagy for pro-viral functions, the precise manner by which this is accomplished remains to be elucidated. In the current manuscript, we sought to address the mechanism by which EVs subvert the autophagy pathway using Coxsackievirus B3 (CVB3) as a model. We showed that CVB3 infection selectively degrades the autophagy cysteine protease ATG4A but not other isoforms. Exogenous expression of an N-terminally Flag-labeled ATG4A demonstrated the emergence of a 43-kDa cleavage fragment following CVB3 infection. Furthermore, bioinformatics analysis coupled with site-directed mutagenesis and in vitro cleavage assays revealed that CVB3 protease 2A cleaves ATG4A before glycine 374. Using a combination of genetic silencing and overexpression studies, we demonstrated a novel pro-viral function for the autophagy protease ATG4A. Additionally, cleavage of ATG4A was associated with a loss of autophagy function of the truncated cleavage fragment. Collectively, our study identified ATG4A as a novel substrate of CVB3 protease, leading to disrupted host cellular function and sheds further light on viral mechanisms of autophagy dysregulation.

## 1. Introduction

Enteroviruses (EVs) are a genus of small, non-enveloped, single-stranded positive RNA viruses of the Picornaviridae family and include poliovirus, enterovirus A71 (EV-A71), EV-D68, and coxsackievirus B3 (CVB3) [1]. CVB3 in particular is the most prevalent etiological pathogen associated with viral myocarditis, which is the most common cause of heart failure and sudden death in infants, adolescents, and young adults [2,3,4]. Despite significant efforts, no clinically validated treatment for this condition is available.

A typical EV lifecycle begins when the viral capsid structure directly binds to a cell entry receptor such as the coxsackievirus and adenovirus receptor (CAR) for CVB3, resulting in the internalization of the virus [5]. Afterwards, viral RNA is translated into a single large polypeptide, which is subsequently cleaved by two EV-encoded cysteine proteases (2A and 3C) into individual structural and non-structural proteins [5]. In the meantime, the EV-encoded RNA-dependent RNA polymerase, 3D^pol^, replicates the viral genome through a negative strand intermediate to generate positive-stranded viral RNA that is subsequently packaged into icosahedral viral capsids. Eventually, viral progeny is released to infect nearby cells.

EVs are obligate intracellular parasites that rely heavily on the cellular machinery of their hosts to replicate [1]. The macroautophagy (hereafter referred to as autophagy) pathway is one of these cellular machineries that has been extensively explored. As a result, investigation of the connection between the CVB3 lifecycle and autophagy is warranted. Autophagy is an evolutionarily conserved, fundamental cellular homeostatic pathway by which undesired cytosolic entities such as aggregated proteins, damaged organelles and pathogens are sequestered within double-membraned vesicles (autophagosomes) and sent to lysosomes for degradation [6]. The biogenesis of the autophagosome begins with the formation of a phagophore, a lipid bilayer that elongates and encloses with the help of a ubiquitin-like conjugation system (ATG5-ATG12-ATG16L) [7,8]. This complex recruits the ubiquitin-like, microtubule-associated protein 1A/1B-light chain 3 (LC3) proteins to the developing phagophore and facilitates LC3 attachment to phosphatidylethanolamine (PE) lipids resulting in membrane elongation, curvature, and sealing of the phagophore into a mature autophagosome [1,9]. Unlike the sole ATG8 homolog of LC3 in yeast, mammals have two subfamilies: LC3 and γ-aminobutyric acid receptor-associated protein (GABARAP), each encoding three homologs (LC3A, LC3B, LC3C and GABARAP, GABARAPL1, GABARAPL2) [10]. Among these homologs, LC3B is the most studied [9]. The processing of LC3 for lipid conjugation is dependent on another ubiquitin-like system, along with the only protease in the ATG protein family, ATG4 [11,12]. The ATG4 cysteine protease directly cleaves pro-LC3 (the precursor of LC3) to expose a C-terminal glycine residue to generate a pool of cytosolic LC3-I proteins. Subsequently, LC3-I is recruited to autophagic membranes and directly conjugated in a ubiquitin-like manner with PE lipids, a process termed lipidation [11,12]. ATG4 is also responsible for cleaving LC3 from PE, termed delipidation, a process that recycles the cellular pool of LC3 for subsequent autophagosome biogenesis [12]. In short, ATG4 is an essential component of the autophagy process as it serves a central role in the lipidation and delipidation of LC3, during autophagosome biogenesis. Among the four ATG4 homologs that exist in mammals (ATG4A, ATG4B, ATG4C and ATG4D) [13], ATG4B was shown to have the most LC3 processing activity as well as the most studied, followed by ATG4A, whereas ATG4C and ATG4D show little activity towards all subtypes of LC3 [13]. 

The autophagy pathway is a central component of cell-autonomous innate immunity that selectively targets and degrades various invading viral pathogens following infection. Nevertheless, many studies have reported that EVs can subvert the autophagy process as a viral strategy to evade host innate immunity and facilitate enhanced viral propagation. Most major EVs, such as poliovirus, coxsackievirus, enterovirus A71 and D68, have been shown to stimulate autophagosome formation upon infection, as evidenced by LC3 lipidation and puncta formation, as well as the presence of intracellular double-membraned vesicles [14,15]. Intriguingly, recent evidence suggests that EV-induced autophagy is likely a non-canonical process that bypasses the requirements for ULK1/2 and PI3K kinase complexes, components that are essential for the normal autophagy process (e.g., under starvation stress) [16,17].

To better understand how EV subverts autophagy, we screened canonical autophagy factors and found out that several proteins were downregulated following EV infection, including ATG4A [17,18]. In the current study, we sought to clarify the interaction between EV and ATG4A and determine the role of ATG4A during viral infection using CVB3 as a model enterovirus. We discover that ATG4A is a novel target of viral protease 2A and demonstrate a concentration-dependent effect of ATG4A during CVB3 viral propagation. Furthermore, we show that CVB3 cleaves ATG4A to disrupt its canonical autophagy role.

## 2. Materials and Methods

### 2.1. Cell Culture

HeLa and HEK293T cells obtained from American Type Culture Collection (ATCC) were cultured in Dulbecco’s Modified Eagle Medium (DMEM) supplemented with 10% fetal bovine serum and 1% penicillin/streptomycin.

### 2.2. Plasmids

The pEGFP-LC3 plasmid was purchased from addgene (#21073). The 3′Flag-ATG4A (hereby shortened as Flag-ATG4A) and 3′Flag-ATG4B (hereby shortened as Flag-ATG4B) were generated by cloning ATG4A and ATG4B into the multiple cloning sites of p3×Flag CMV10 vector with HindIII/Xbal (ATG4A) and EcoRI/BamHI (ATG4B), respectively. Mutant G374E-ATG4A was generated by site directed mutagenesis and cloning into 3′Flag plasmid. Truncated-ATG4A (deletion at C-terminal after G374) was generated by cloning into 3′Flag plasmid with HindIII and KpnI. Table 1 summarizes the primer sequences used to amplify each gene.

### 2.3. Transfections

Plasmid transfections were performed using Lipofectamine 2000 (11668019, Invitrogen) according to the manufacturer’s guidelines. Briefly, 1 mg of DNA diluted in OPTI-MEM (31985062, Thermo Fisher Scientific) was combined with Lipofectamine and incubated at room temperature for 15 min. DNA/lipofectamine mixture was then added dropwise to 6-well plates along with cells (5.5 × 10^5^) for transfection overnight. For RNA interference transfection, Hela cells were transiently transfected with ATG4A siRNA (sc-91197, Santa Cruz Biotechnology) and scrambled siRNA (sc-37007, Santa Cruz Biotechnology) using lipofectamine 2000 following the manufacturer’s instruction. To assess cell viability, siRNA-treated cells were incubated with MTS (3-(4,5-dimethylthiazol-2-yl)-5-(3-carboxymethoxyphenyl)-2-(4-sulfophenyl)-2H-tetrazolium) reagent (Abcam, cat#ab197010) following the manufacturer’s guidelines. Briefly, cells were incubated with 10% MTS solution in complete medium for 2 h at 37 °C. SpectraMax iD3 microplate reader was used to record absorbance at 490 nm of control and treated cells.

### 2.4. Viral Infections

Seeded cells were infected with CVB3 at different multiplicities of infection (MOI) for various time points as indicated or sham-infected with DMEM. HeLa cells were infected with an MOI = 10 whereas HEK293T cells were infected with an MOI = 100 as previously reported [17]. Viral titers were quantified as previously described [19]. Briefly, culture medium from infected samples was serially diluted and transferred to a bed of HeLa cells in 60-well Terasaki plates. After 48 h incubation, 50% tissue culture infective dose titer (TCID50) was calculated by the statistical method of Reed and Muench (HM., 1938 #53). Viral titers are expressed as plaque-forming unit (PFU)/mL with 1 infectious unit equal to 0.7 TCID50.

### 2.5. Western Blot Analysis

Cells were lysed and harvested with Modified Oncogene Science lysis buffer (MOSLB, 10 mM HEPES with a pH 7.4, 50 mM Na pyrophosphate, 50 mM NaF, 50 mM NaCl, 5 mM EDTA, 5 mM EGTA, 100 µM Na3VO4, 1% Triton X-100) containing protease inhibitors. Lysates were denatured in 6× sodium dodecyl sulfate protein loading buffer (62.5 mM Tris-HCl at the pH of 6.8, 2% (*w*/*v*) SDS, 10% glycerol, 0.01% (*w*/*v*) bromophenol blue, and 1.25 M of dithiothreitol). The denatured proteins were then separated by sodium dodecyl sulfate-polyacrylamide gel electrophoresis. Proteins were transferred onto 0.45 μm nitrocellulose membrane for 80 min. at 100 V. For total protein visualization, membranes were stained with Ponceau S (0.1% *w*/*v* in 5% acetic acid) and visualized before rinsing with TBST wash buffer. After protein transfer, the membranes were incubated with primary antibodies at 4 °C overnight. The following primary antibodies were used: anti-ATG4A antibody (CST#7613, monoclonal antibody targeting C-terminus, kindly provided by Dr. Sharon Gorski, anti-ATG4B antibody (A2981, Sigma-Aldrich, polyclonal antibody targeting amino acids 6–22), anti-ATG4C antibody (5262, Cell Signaling Technology, polyclonal antibody targeting peptide surrounding Ser430), anti-VP1 antibody (M47, Mediagnost), anti-β-actin antibody (sc-47778, Santa Cruz Biotechnology), anti-LC3B antibody (NB100-2220, Novus Biologicals), anti-Flag antibody (F1804, Sigma-Aldrich), anti-HSP90 (MA110372, Thermofisher). The membranes were then incubated with secondary antibodies for 1 h at room temperature before visualization by chemiluminescence. All primary and secondary antibodies were diluted in 2.5% BSA + Tris-buffered saline with 0.1% Tween^®^ 20 detergent (TBST) with 1:1000 dilutions.

### 2.6. In Vitro Cleavage Assay

Cell lysates overexpressing Flag-ATG4A were incubated with purified wild-type (WT) CVB3 protease 2A or 3C (0.1 µg) in a cleavage assay buffer (20 mM HEPES pH 7.4, 150 mM potassium acetate, and 1 mM DTT) for 2 h at 37 °C. Reactions were terminated with 6× SDS sample buffer, followed by 95 °C denaturation and subsequent Western blot analysis.

### 2.7. Real Time Quantitative PCR

Total RNA was extracted using RNeasy Mini Kit (74104, Qiagen). To determine viral gene expression levels, quantitative PCR (qPCR) was conducted with primer pairs targeting viral 2A (forward primer: 5′-GCT TTG CAG ACA TCC GTG ATC-3′; reverse primer: 5′-CAA GCT GTG TTC CAC ATA GTC CTT CA-3′), VP1 (forward primer: 5′-ACA TGG TGC GAA GAG TCT ATT GAG-3′; reverse primer: 5′-TGC TCC GCA GTT AGG ATT AGC-3′), ATG4A (forward primer: 5′-CCA AGC CAG AAG TGA CAA CCAC-3′; reverse primer: 5′-GAC AGA CCT TCA AGT TGA GTT CC-3′) and ACTB (forward primer: 5′-ACT GGA ACG GTG AAG GTG AC-3′; reverse primer: 5′-GTG GAC TTG GGA GAG GAC TG-3′). The qPCR reaction containing 1 μg of RNA was performed using the TaqMan™ RNA-to-CT™ 1-Step Kit (4392653, Thermo Fisher Scientific) on a ViiA 7 Real-Time PCR System (Applied Biosystems). Samples were run in quadruplicate and analyzed using comparative CT (2^−ΔΔ*C*T^) method with control samples and presented as relative quantitation (RQ). Gene expression of 2A and VP1 was normalized to ACTB mRNA.

### 2.8. Immunoprecipitation

Immunoprecipitation was performed using anti-FLAG magnetic beads (M8823 Sigma-Aldrich). Briefly, cell lysates were incubated with the beads overnight at 4 °C. Bounded proteins were then eluted with 1% SDS sample buffer and the reactions were terminated by incubating at 95 °C for 10 min. Immunoprecipitated samples were subjected to Western blot analysis.

### 2.9. Confocal Microscopy

HeLa cells were cultured in 8-well chamber slides (Labtek, 155411) for 24 h. The cells were then transfected with pEGFP-LC3 together with WT-ATG4A or T-ATG4A for an additional 24 h. GFP-LC3 puncta was visualized by Zeiss LSM 880 inverted confocal microscopy and quantified using NIH Image J version 1.53c. Number of GFP-LC3 puncta-positive cells was counted and divided by total cell number to yield the percentage of GFP-LC3 puncta-positive cells (*n* > 30 cells per condition).

### 2.10. Statistical Analysis

Densitometric analysis of Western blots was performed by quantifying the intensity of protein bands and normalizing them to ACTB band of sham group unless otherwise noted, using NIH Image J version 1.53c. Each experiment had 3 biological replicates unless otherwise specified. Unpaired student *t*-test was used for two-group comparison. For multiple group comparisons, one way ANOVA was performed to determine the statistical significance with Tukey’s post-hoc test. Standard deviation was used and *p* < 0.05 was considered to be statistically significant.

## 3. Results

### 3.1. CVB3 Selectively Cleaves ATG4A but Not ATG4B during Late Infection

To study how CVB3 interacts with different homologs of ATG4, HEK293T and HeLa cells were infected with CVB3 at a multiplicity of infection (MOI) of 100 and 10, respectively, for various time-courses. The viral doses selected have been previously shown to facilitate optimal infection in respective cell type [17]. Cell lysates were collected and subjected to Western blot analysis probing for ATG4A, ATG4B, and ATG4C. Viral capsid protein VP1 was probed as a control for infection, and ACTB was probed as a loading control. Of note, cell lysates for CVB3-infected HEK293 and HeLa cells were harvested at time-points preceding virus-induced apoptosis (observed at 24 h and 9 h post-infection (pi), respectively, for HEK293T and HeLa cells. As shown in Figure 1A,B, the level of ATG4A decreased after the effective replication of CVB3 (depicted by the appearance of VP1 band) at 8 h pi in HEK293T cells and 5 h pi in HeLa cells, suggesting that the decrease of ATG4A could be mediated by viral protein production. In contrast, CVB3 infection did not cause levels of ATG4B and ATG4C to decrease (Figure 1A,B). Furthermore, we tested the gene expression levels of ATG4A in sham and CVB3-infected cells and confirmed that gene expression was unaltered between sham and CVB3 at a timepoint of infection (7 h pi) when ATG4A protein was declining (Figure 1C).

We then decided to elucidate the mechanism of degradation. Since no endogenous ATG4A fragments were observed after CVB3 infection with an anti-C-terminal targeting antibody, we generated two constructs by inducing Flag tags to the N terminus of ATG4A and ATG4B. ATG4B was chosen to be cloned as it is the most active isoform of ATG4 [20]. Flag-ATG4A was transfected into HeLa cells followed by a time-course of CVB3 infection. Cell lysates were collected and probed with anti- Flag antibody to detect exogenous Flag-labelled ATG4A as well as anti-VP1 and anti-ACTB as viral infection and loading controls, respectively. Western blot analysis revealed a fragment below the normal molecular weight of full-length ATG4A at 5 h pi, suggesting a possible cleavage (Figure 1D). This appearance of the cleavage band at 5 h is consistent with the degradation of ATG4A shown in Figure 1B. To verify that this cleavage was specific to ATG4A, HEK293T cells were transiently transfected with Flag-ATG4A and Flag-ATG4B, followed by sham or CVB3 infection for 8 h. Indeed, the cleavage was specific to ATG4A as this cleavage was not observed in cells expressing Flag-ATG4B (Figure 1E). Collectively, we showed that CVB3 differentially modulates ATG4 homologs through proteolytic activities. 

### 3.2. ATG4A Is Cleaved by Viral Protease 2A before Glycine 374

Given that CVB3 infection resulted in the proteolytic processing of ATG4A, we next sought to identify whether viral protease is responsible for ATG4A cleavage. CVB3 expresses two cysteine proteases, 2A and 3C, that are used primarily to process the viral polyprotein. We performed an in vitro cleavage assay, in which Flag-ATG4A was transfected into HEK293T cells and the corresponding lysates were incubated with either purified 2A or 3C. Western blot analysis demonstrated no evident changes in the levels of full-length Flag-ATG4A in 3C-treated cell lysates (Figure 2A, left panel). As a positive control for protease activity, SNAP29, a previously reported substrate of 3C, was assessed for cleavage efficacy [21]. In contrast, when lysates were treated with purified 2A, there was a significant decrease in the intensity of Flag-ATG4A band as well as the appearance of cleavage band for elf4G (a protein known to be cleaved by EV 2A [22]), suggesting that 2A is responsible for cleaving ATG4A (Figure 2A, middle panel). However, no cleavage fragments of WT-ATG4A were shown, which could potentially be explained by the strong activity of purified 2A in the in vitro setting that completely degrades ATG4A into undetectable fragments (Figure 2A). Given the significant alteration in ACTB protein expression following 2A treatment, we further assessed the total protein levels following in vitro cleavage using Ponceau S stain as well as an additional housekeeping protein, heat shock protein 90 (HSP90) (Figure 2A, right). Collectively, these additional stains revealed no significant differences between control and 2A-treated lysates, suggesting that ACTB may be a specific target.

We next set out to identify the site of cleavage by 2A. The consensus cleavage sequence of protease 2A, shown in Figure 2B, has a highly conserved scissile bond between P1 and the glycine (P1′) residue. To identify the precise location of cleavage, we screened the ATG4A open reading frame and cross-referenced with the consensus cleavage sequence. Five potential cleavage sites were identified, and the molecular size of each Flag-tagged cleavage fragments was estimated via Expasy compute pl/Mw tool (Figure 2B). However, only one of the putative cleavage sites, glycine (G) 374, was consistent with the size of the cleavage fragment shown in Figure 1C,D. To verify whether G374 is the cleavage site of ATG4A, we mutated the glycine at residue 374 to glutamic acid (E) via site-directed mutagenesis, and transfected the mutant constructs into HEK293T cells along with vector and WT-ATG4A as controls for 24 h. Following CVB3 infections, it was found that WT-ATG4A was cleaved while the G374E mutant of ATG4A was not, supporting our hypothesis that CVB3 cleaves ATG4A before the G374 residue (Figure 2C). The ATG4A-specific cleavage generates a truncated fragment of approximately 43 kDa (lower band) just below the full-length 50 kDa protein (upper band) when probing with N-terminal anti-FLAG antibody. A schematic illustration of ATG4A protein and its respective functional domains is provided in Figure 2D as well as the C-terminal amino acid sequences of ATG4A and ATG4B highlighting the divergence at glycine 374. The cleavage of ATG4A by 2A results in a deletion of the C-terminal LC3 interacting region (LIR) (Figure 2D).

### 3.3. Endogenous Concentration of ATG4A Has Pro-Viral Function

Given that CVB3 directly cleaves ATG4A, we wondered what precise role this host factor plays during the life cycle of CVB3 infection. To address this, we silenced endogenous ATG4A by RNA interference in HeLa cells for 48 h, followed by CVB3 infection for an additional 8 h. Cell lysates were harvested and subjected to Western blot analysis to assess the efficacy of ATG4A silencing and expression level of viral capsid protein VP1. The latter was used as a proxy to measure viral replication. We found that knockdown of ATG4A was associated with significantly reduced viral protein production (Figure 3A). Consistent with this observation, ATG4A-silenced cells also showed decreased quantity of viral RNA (as measured with viral 2A primers) and viral titers compared to cells treated with control siRNA, suggesting a pro-viral function for ATG4A (Figure 3B). As a control, we assessed the viability of cells following 48 h silencing of ATG4A and observed no significant cell death as compared to siCON-treated cells (Figure 3C). Surprisingly, we found that overexpression of exogenous Flag-ATG4A significantly reduced VP1 production compared to vector control, suggesting that a high concentration of ATG4A exerts anti-viral effect (Figure 3D). To assess the functional significance of ATG4A overexpression, we evaluated the processing of LC3-I to LC3-II, a marker of autophagy induction. Consistent with previous findings [17], the LC3-II to LC3-I ratio was significantly enhanced following CVB3 infection (Figure 3D,E). In contrast, ATG4A overexpressed samples showed significant decrease in LC3-II/LC3-I following CVB3 infection, possibly indicating promotion of LC3-II delipidation back to LC3-I, which disrupts autophagy (Figure 3D,E).

Given that ATG4A is cleaved by viral protease 2A before glycine 374, we generated a plasmid expressing T-ATG4A (lacking amino acids 374–398) to mimic the cleaved fragment. We then assessed the effect of overexpression of WT-ATG4A and T-ATG4A on viral RNA expression. HeLa cells were transiently transfected with vector control, WT-ATG4A, or T-ATG4A. Following 24 h, cells were subjected to CVB3 infection and RNA was harvested at 1, 4, and 8 h pi. Intriguingly, quantitative assessment of viral RNA using viral VP1 primers demonstrated no significant changes between control, WT-ATG4A, and T-ATG4A expressing cells, suggesting that both WT-ATG4A and T-ATG4A do not significantly perturb viral RNA production (Figure 3F). We used both viral genes 2A (Figure 3B) and VP1 (Figure 3F) to measure viral RNA because CVB3 utilizes a single open reading frame encoding a single-copy of each viral gene. This unique characteristic of monopartite viruses such as CVB3 allows for the accurate quantitation of viral RNA irrespective of which viral gene is used for RT-qPCR.

Overall, these results may suggest a complex role of ATG4A in CVB3 propagation whereby both (1) loss of endogenous ATG4A and (2) overexpression of exogenous ATG4A are associated with anti-viral effects. We then wish to gain more mechanistic insight by exploring the functional impact of the cleavage of the ATG4A.

### 3.4. Truncated ATG4A Has Impaired Functional Capacity

We next asked whether T-ATG4A exhibits any impairment in protein function. Since the cleavage of ATG4A before glycine 374 results in the loss of the LIR on the C-terminus, we therefore rationalized that T-ATG4A may have impaired autophagy function due to disrupted LC3/GABARAP interaction. To assess whether there are any potential functional deficits of T-ATG4A with respect to autophagy, HeLa cells were transiently transfected with either WT-ATG4A or T-ATG4A along with GFP-labelled LC3. After 24 h, confocal imaging revealed a significantly increased accumulation of GFP-LC3-puncta in T-ATG4A expressing but not in WT-ATG4A expressing cells (Figure 4A). The accumulation of the LC3 puncta could be due to a decrease and inefficiency in the LC3-II delipidation process, inferring that CVB3 is retaining autophagosome membrane and indirectly preventing lysosomal fusion. In addition to LC3, ATG4 family proteins also regulate the processing of GABARAPs (homologs of LC3) and previous research suggests that ATG4A has high binding and processing affinity toward GABARAPs [13]. To further assess the functional impairment of T-ATG4A, we performed a co-immunoprecipitation assay to evaluate the interaction between either WT-ATG4A, WT-ATG4B, or T-ATG4A and GABARAP. As expected, WT-ATG4A, rather than WT-ATG4B, preferentially interacted with GABARAP (Figure 4B). Rationally, T-ATG4A had a diminished capacity to interact with GABARAP, an observation that is consistent with the loss of LIR domain following CVB3-mediated cleavage (Figure 4B). Collectively, this evidence demonstrates that virally mediated cleavage of ATG4A results in impaired LC3 and GABARAP regulation leading to disrupted autophagy. Finally, we sought to test whether ATG4A mutants had direct impact on viral protein production. To this end, HeLa cells were transiently transfected with constructs encoding either WT, truncated, or non-cleavable mutant ATG4A. Interestingly, no significant alterations in viral protein production was observed among these three constructs despite an overall downtrend as compared to vector control, suggesting an overall anti-viral role of ATG4A (Figure 4C).

## 4. Discussion

Despite a miniscule armada of merely eleven viral proteins, EVs are capable of disrupting complex biological systems, such as the cardiovascular and central nervous system, leading to a broad spectrum of human illnesses as minor as the common cold or as severe as viral myocarditis and heart failure [3,23]. Among EVs’ arsenals, two virally encoded cysteine proteases, 2A and 3C, are emerging as extremely pathogenic viral factors that disrupt important cellular pathways. The 2A and 3C primarily function to process the viral polyprotein into individual viral proteins or intermediates; however, these proteases routinely target host factors as a strategy to facilitate viral propagation and/or evade host defenses [24,25]. Many host substrates of EV proteases have been identified, including several candidates within the autophagy pathway. For example, the autophagy initiating serine-threonine kinase, unc-51-like kinase (ULK1), was recently shown to be targeted by CVB3 to disrupt canonical autophagy signaling. It was demonstrated that viral protease 3C cleaves ULK1 after a central glutamine (Q) 524 residue, resulting in the separation of the N-terminal kinase domain from the C-terminal substrate interaction region and consequently impairing host autophagy regulation in favor of viral pathogenesis [17]. Therefore, we hypothesized that CVB3 proteases are able to target other autophagy related proteins to further hinder autophagic processes. 

Our results demonstrated that CVB3 infection led to the specific downregulation of ATG4A. We discovered that degradation of ATG4A is the consequence of virus-encoded protease 2A-mediated cleavage. The cleavage takes place before G374, which separates the C-terminal kinase domains from the N-terminal LIR. In this study, we focused on the N-terminal cleavage fragment (~43 kDa) as the C-terminal fragment is too small (<3 kDa) to detect by conventional Western blot analysis. Although it is possible that ATG4A may be a collateral target of the promiscuous 2A protease, findings from the current study clearly demonstrate an impaired functional autophagy through abnormalities in LC3 lipidation upon the expression of cleaved T-ATG4A supporting a potential strategy by which virus may disrupt canonical autophagy functions through direct targeting of autophagy factors.

Given that CVB3 targets ATG4A, we then wondered what role ATG4A plays in CVB3 infection. Autophagy can have either pro-viral or anti-viral effect on diverse viruses. For example, Hait et al. [26] reported that missense mutations in genes encoding ATG4A and LC3B2 disrupt autophagy, resulting in increased herpes simplex virus 2 replication and susceptibility to viral infection in both primary fibroblasts and a neuroblastoma cell line, highlighting the importance of the anti-viral effect of autophagy. Conversely, EVs, such as poliovirus, were shown to usurp autophagy for pro-viral functions, likely by recruiting host (autophagosome) membrane for replication [27]. In our study, endogenous ATG4A was shown to facilitate viral propagation. We observed both reduced viral RNA and protein expression following ATG4A knockdown, suggesting the necessity of ATG4A presence in viral growth, yet its precise mode of action remains unknown. We wondered why CVB3 would cleave a pro-viral protein. One possibility is that the timing of cleavage could play a role given that ATG4A processing began 5 h post infection, a time point that is considered to be late stage of the viral lifecycle in HeLa cells [28]. Given that viral-induced double membrane replication organelles appear early, it is plausible that non-cleaved ATG4A exerts its pro-viral effect at an early stage as it facilitates autophagosome biogenesis that can serve as replication compartments for CVB3 [28,29]. Additionally, the pro-viral role of ATG4A may be autophagy independent. One possibility is that CVB3 may co-opt the cysteine protease function of ATG4A, similar to the virally encoded cysteine proteases 2A and 3C, to ultimately support viral maturation. 

Previous studies have shown that pro-viral factors can be targeted by viral proteases during late infection [17,19]. Interestingly, overexpression of exogenous ATG4A is associated with significant reduction in viral protein production although no changes in viral RNA synthesis were observed, signifying that this overexpressed anti-viral function of ATG4A is not viral RNA dependent. Instead, higher concentrations of ATG4A could be acting upon viral proteins or has an effect on other cellular pathways that affect viral protein synthesis (off-target effect). Interestingly, the transfection of a non-cleavable mutant (G374E) or truncated ATG4A, did not significantly impact viral protein production as compared to wildtype ATG4A expression although all three constructs showed a downtrend in viral protein compared to vector only transfected cells, suggesting a general anti-viral role of ATG4A mutants used in the current study. Of note, overexpression of ATG4A significantly blocks CVB3-induced lipidation likely through its delipidation function. We posit that the anti-viral effects of exogenous ATG4A may be the result of its delipidation function and a primary reason why viral protease 2A may disrupt this function through ATG4A cleavage. Additionally, a large amount of exogenous ATG4A may suppress viral replication by potentially accelerating the recycling, or delipidation, of LC3-II. Taken together, these divergent observations suggest that ATG4A may have pro-viral functions during the early stages but anti-viral effects during late infection, providing a plausible explanation of viral-mediated cleavage of ATG4A during late infection.

In this study, we showed that the virally mediated cleavage of the LIR domain of ATG4A leads to impaired autophagy function, including reduced LC3 delipidation and GABARAP interaction. Interestingly, the overexpression of WT-ATG4A is associated with a significant reduction in LC3 puncta whereas approximately 70% of cells expressing T-ATG4A exhibit accumulation of LC3 puncta. Increased LC3 puncta can be interpreted as either an enhancement in the autophagosome biogenesis process or a disruption of the autophagic recycling machinery, resulting in reduced clearance of LC3. We rationalize that ATG4A overexpression is likely enhancing the delipidation process whereas overexpression of T-ATG4A results in the accumulation of LC3 puncta due to an impaired LC3-II delipidation process. The truncation of ATG4A may be a viral strategy to retain autophagosome membranes and indirectly prevent lysosomal fusion. Indeed, EVs have previously been shown to utilize viral protease to disrupt the autophagosome-lysosome fusion process [21,30]. In the current study, we utilized a Flag-tagged ATG4A, and we cannot exclude the possibility that 3×Flag tag has other impacts. However, in our previous research, the use of a 3×Flag tag did not interfere with protein function likely due to the small molecular size (~3 kDa) of the tag [21].

In addition to impaired LC3 interaction, we also showed decreased physical association between T-ATG4A and GABARAP. ATG4A has been reported to have high affinity and catalytic efficiency toward GABARAP, which was confirmed in our interaction assay [13]. Notably, GABARAP is preferentially involved in membrane tethering and fusion [10]. It is therefore postulated that cleavage of ATG4A may hinder the ability of GABARAP to promote autophagosome-lysosome fusion, and consequent viral protein degradation.

In summary, our study identified ATG4A as a novel pro-viral host factor that is co-opted by EV protease to impair host autophagy. To our knowledge, this is the first reported cleavage of a host protease by a *Picornaviridae* viral protease. The cleavage poses one of the underlying mechanisms by which EVs evade autophagy. 

## Figures and Tables

**Figure 1 viruses-14-02026-f001:**
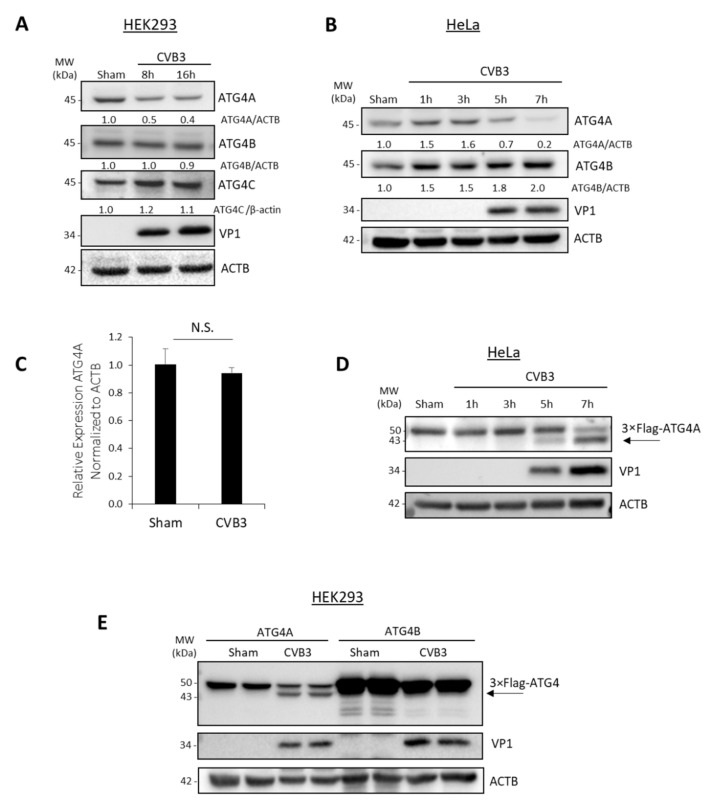
ATG4A is cleaved following CVB3 infection. (**A**,**B**) HEK293T (**A**) and HeLa (**B**) cells were infected with CVB3 at an MOI of 100 and 10, respectively, for various time as indicated or sham-infected with DMEM. Cell lysates were harvested for Western blot analysis of ATG4A, ATG4B, ATG4C. Viral capsid protein VP1 and housekeeping protein ACTB were probed and served as viral infection and loading control, respectively. Densitometric analysis of ATG4A, ATG4B, and ATG4C protein levels was performed using NIH Image J, normalized to ACTB and presented underneath each band as fold changes where sham groups were arbitrarily set to 1.0. (**C**) Sham and CVB3-infected HeLa cells were harvested for RNA purification, followed by RT-qPCR assessment of ATG4A gene and normalized to housekeeping gene ACTB (*n* = 3, mean ± S.D.) (**D**) HeLa cells were transiently transfected with construct encoding Flag-ATG4A for 24 h and subsequently infected with CVB3 (MOI = 10) for the indicated timepoints. Cell lysates were harvested for Western blot analysis with anti-Flag antibody. (**E**) HEK293T cells were transfected with 3×Flag-ATG4A or 3×Flag-ATG4B plasmids as indicated for 24 h, followed by sham or CVB3 infection (MOI = 100) for 8 h. Cell lysates were harvested and probed for Flag-tag, VP1 and ACTB. Arrows indicate the cleavage fragments. Results in this figure are representative of two to three independent experiments.

**Figure 2 viruses-14-02026-f002:**
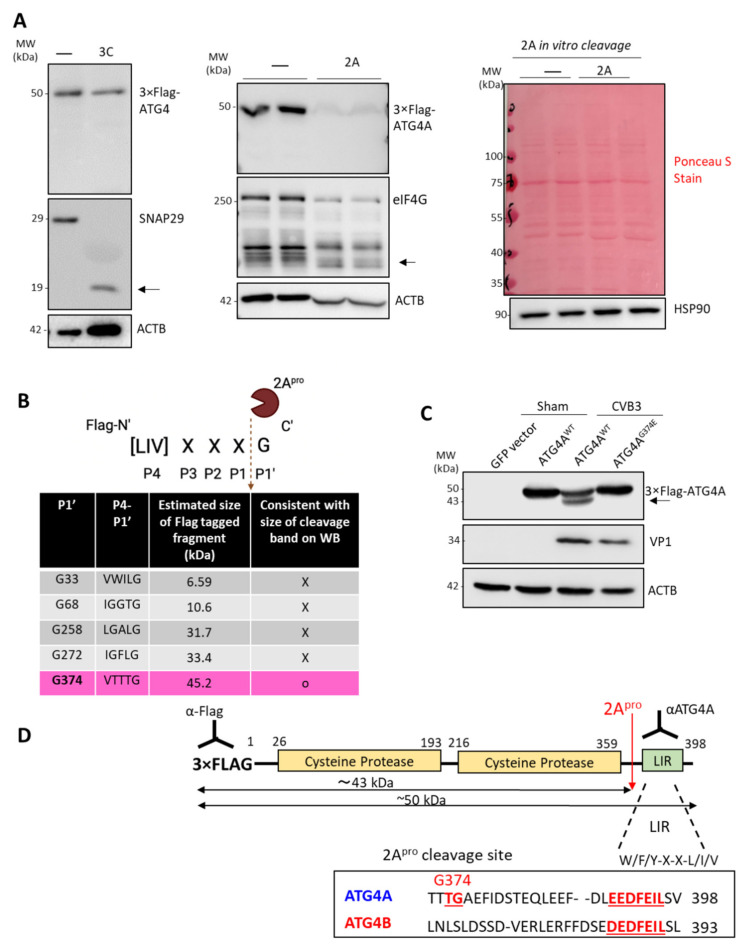
ATG4A is cleaved by viral protease 2A before glycine 374. (**A**) In vitro cleavage assay was performed with no protease control, purified viral protease 3C (**left**) or 2A (**middle**) in the presence of Flag-ATG4A. SNAP29 and eIF4G were used as a positive control for 3C and 2A activity, respectively. Membranes from 2A in vitro cleavage (**middle**) were stained with Ponceau S Stain or immunostained with anti-HSP90 antibody to probe total protein and alternative loading control respectively. (**B**) Illustration of the potential cleavage sites in the open reading frame of ATG4A. Estimated size of Flag-tagged fragments was calculated by adding 3×Flag (2.7 kDa) to size of potential N-terminal cleaved fragments (estimated via Expasy Compute pl/Mw tool). Predicted fragment size consistent with cleavage is highlighted (**C**) HEK293T cells were transfected with GFP, ATG4^WT^, or ATG4A^G374E^ for 24 h, followed by sham or CVB3 infection (MOI = 100) for 8 h. Cell lysates were harvested and probed for FLAG-ATG4A using anti-Flag antibody, VP1 and ACTB. (**D**) Schematic diagram of the function domains of Flag-ATG4A, identified cleavage site by 2A, and the sizes of cleavage fragments. Sequence alignments between ATG4A and ATG4B within the cleavage region of the fragments are shown. LIR: LC3-interacting region is shown in underlined red text. Results in this figure are representative of two to three independent experiments.

**Figure 3 viruses-14-02026-f003:**
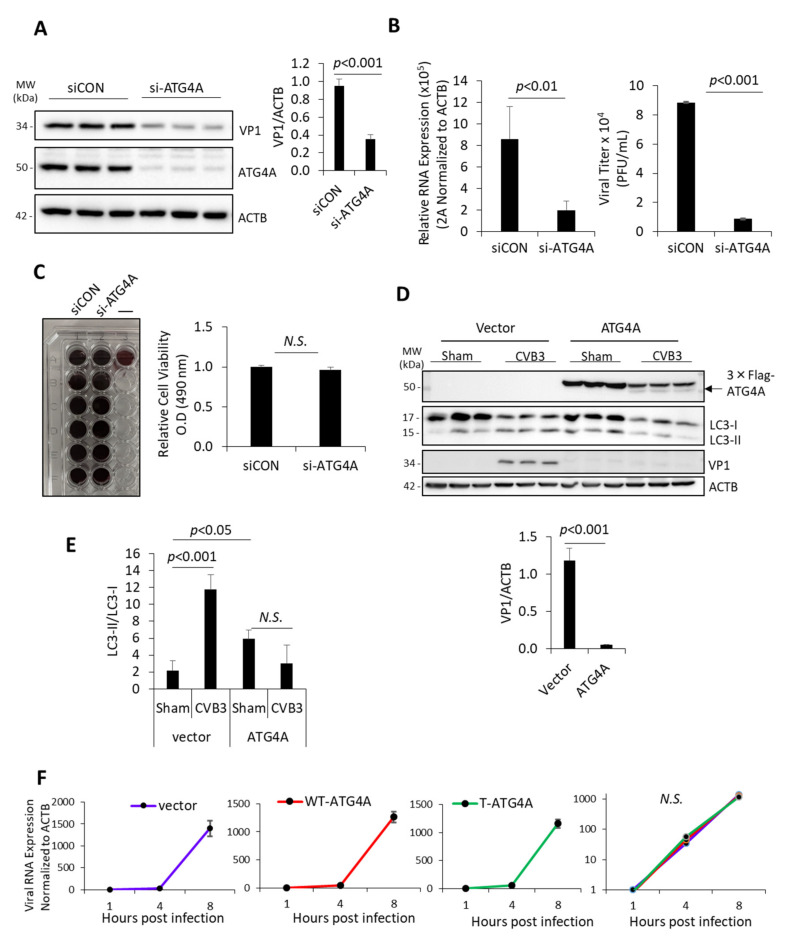
Endogenous Concentration of ATG4A has pro-viral function. (**A**,**B**) HeLa cells were transfected with control siRNA (siCON) or siRNA specifically targeting ATG4A for 48 h, followed by CVB3 (MOI = 10) or sham infection for 8 h. Western blot was conducted to analyze VP1 expression and verify knockdown efficacy of ATG4A (A, left). Level of VP1 was quantified by densitometry using Image J and presented as mean ± S.D., *n* = 3 (**A**, **right**). (**B**) RT-qPCR (**left**) and TCID50 assessment (**right**) were performed to determine viral RNA expression and viral titers in siCON- or si-ATG4A-treated cells and presented as mean ± S.D., *n* = 3. Statistical difference was determined by unpaired Student’s *t*-test. (**C**) Cell viability was assessed following 48 h treatment of either siCON or si-ATG4A using MTS assay and presented as relative cell viability (mean ± S.D., *n* = 6) (**D**,**E**) HEK293T cells were transfected with empty vector or Flag-ATG4A for 24 h. Cells were then infected with CVB3 (MOI = 100) or sham-infected for 8 h. Cell lysates were harvested and probed for Flag-ATG4A (with anti-Flag antibody), LC3, and ACTB (**E**, **left**). Densitometric analysis of VP1 (**D**, **right**) and LC3-II (**E**) protein levels was conducted via NIH Image J, normalized to ACTB and LC3-I, respectively and presented as mean ± S.D., *n* = 3. Statistical analysis of VP1 and LC3-II/LC3-I densitometry was performed by student *t*-test and ANOVA with Tukey post-hoc test respectively (**F**) HeLa cells transfected with either vector, wildtype (WT)- Flag-ATG4A, or truncated (T)-Flag-ATG4A for 24 h, followed by CVB3 infection (MOI = 10) for the indicated time-points. qPCR was performed with primers targeting VP1, normalized to ACTB gene, and presented as mean ± S.D., *n* = 3. Results in this figure are representative of three independent biological experiments.

**Figure 4 viruses-14-02026-f004:**
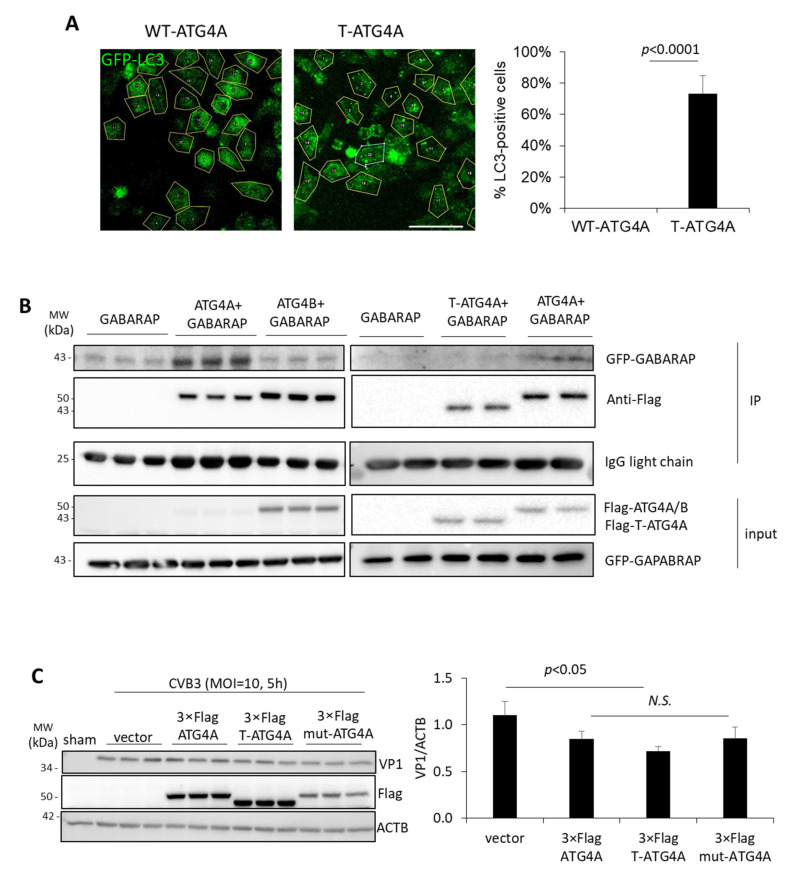
Truncated ATG4A has impaired functional capacity. (**A**) HeLa cells were transfected with GFP-LC3 and either wild-type WT-Flag-ATG4A or T-Flag-ATG4A for 24 h. GFP-LC3 puncta was quantified and presented in right panel as percentage of LC3-puncta positive cells (mean ± S.D., *n* ≥ 30 cells per condition). Scale bar denotes 10 μm. Statistical analysis was performed by unpaired Student’s *t* test. (**B**) HEK293T cells were transfected with GFP-GABARAP alone or in combination with WT-Flag-ATG4A, WT-Flag-ATG4B, or T-Flag-ATG4A. Co-immunoprecipitation (IP) was performed with anti-FLAG antibody and subjected to Western blot analysis. IgG light chain was probed as IP loading control. Results in this figure are representative of two to three independent experiments. (**C**) HeLa cells were transfected with either vector control or constructs encoding WT-3×Flag-ATG4A, truncated 3×Flag-T-ATG4A, or 3×Flag-mut-ATG4A for 24 h prior to CVB3 infection (MOI = 10, 5 h). Lysates were subjected to Western blot analysis for anti-VP1, Flag, and ACTB loading control and densitometry quantified on right panel (mean± S.D., *n* = 3).

**Table 1 viruses-14-02026-t001:** Plasmid names and corresponding primer sequences.

Plasmid	Forward Primer	Reverse Primer
**3×Flag-ATG4A**	5′ AAG CTT (HindIII) ATG GAG TCA GTT TTA TCC AAG TAT GA 3′	5′ TCT AGA (XbaI) CTA CAC ACT CAG AAT CTC AAA ATC TTC 3′
**3×Flag-ATG4B**	5′ GAA TTC (EcoRI) CAT GGA CGC AGC TAC TCT GAC 3′	5′ GGA TCC (BamHI) TCA AAG GGA CAG GAT TTC AA 3′
**G374E-ATG4A**	5′ GAA GTG ACA ACC ACT GAG GCA GAA TTC ATT GAC 3′	5′ GTC AAT GAA TTC TGC CTC AGT GGT TGT CAC TTC 3′
**Delta-G374-ATG4A**	5′ AAG CTT (HindIII) ATG GAG TCA GTT TTA TCC AAG TAT GA 3′	5′ GGT ACC (KpnI) TTA AGT GGT TGT CAC TTC TGG CTT G 3′

## Data Availability

The raw data of this paper is available upon request.

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
