# Peer review of "Coxsackievirus Protease 2A Targets Host Protease ATG4A to Impair Autophagy"

_viruses, 2022, doi:10.3390/v14092026_

Round 1

Reviewer 1 Report

This article by Fan et al entitled, "Coxsackievirus Protease 2A Targets Host Protease ATG4A to 2 Impair Autophagy" describes how CVB 2A protease degrades ATG4A and thereby limits autophagy during CVB infection. This is a short article that is straightforward and the data are generally clear. Further characterizing the degradome of EV proteases is important for our understanding of the viral replication cycle and to help identify potential therapeutic targets. This is a well-written article, but my enthusiasm is tempered by the fact that this is better suited as a brief "Communication" type of article. I have some other major critiques listed below, but generally think the findings presented here are of potential significance for the field.

Major comments:

Much of the characterization of ATG4A elimination is done with western blots to identify reductions in total protein amount and to try to find degradation products. I would imagine there is a good chance that this could partially be described by cell death occurring via treatment with the fairly promiscuous 2A protease. This could also explain why loading controls are also drastically reduced. The authors should clearly indicate what the cell viability is during the time points they investigated. If the cells are apoptotic or necrotic due to overexpression, then changes in protein dynamics could be explained by that. 

The authors briefly mention that it is odd that one of the proteolytic targets is ATG4A, even though autophagy is widely considered a proviral process for these types of viruses. They do explain that this may have more to do with timing, but I feel this explanation is somewhat weak. Further elucidating why turning off autophagy late in infection could be beneficial would enhance the narrative overall. It has been reported that autophagy/flux is important for viral maturation early on, but later in the viral life cycle, autophagic flux seems to be detrimental for the virus (ie-extracellular vesicle release is impaired). Perhaps there might be something there that could explain this phenomenon. Otherwise, again, 2A is very promiscuous, so ATG4A could just be collateral damage.

Author Response

Reviewer #1

This article by Fan et al entitled, "Coxsackievirus Protease 2A Targets Host Protease ATG4A to 2 Impair Autophagy" describes how CVB 2A protease degrades ATG4A and thereby limits autophagy during CVB infection. This is a short article that is straightforward and the data are generally clear. Further characterizing the degradome of EV proteases is important for our understanding of the viral replication cycle and to help identify potential therapeutic targets.

Major comments:

This is a well-written article, but my enthusiasm is tempered by the fact that this is better suited as a brief "Communication" type of article. I have some other major critiques listed below, but generally think the findings presented here are of potential significance for the field.

Response: We thank the reviewer for the positive comments and agree that this article can be published as a communication which we will work with the editors to accomplish.

Much of the characterization of ATG4A elimination is done with western blots to identify reductions in total protein amount and to try to find degradation products. I would imagine there is a good chance that this could partially be described by cell death occurring via treatment with the fairly promiscuous 2A protease. This could also explain why loading controls are also drastically reduced. The authors should clearly indicate what the cell viability is during the time points they investigated. If the cells are apoptotic or necrotic due to overexpression, then changes in protein dynamics could be explained by that. 

Response: We thank the reviewer for this insightful comment. We clarified the reviewer’s concerns regarding cell viability in the revised text. “Cell lysates for CVB3-infected HEK293 and HeLa cells were harvested at timepoint preceding virus-induced apoptosis (observed at 24h and 9h post infection respectively for HEK293T and HeLa cells.” (Page 5). Additionally, we addressed the reviewers comments regarding the reduced ACTB levels during 2A in vitro cleavage. “Given the significant alteration in ACTB protein expression following 2A treatment, we further assessed the total protein levels following in vitro cleavage using Ponceau S stain as well as an additional housekeeping protein, heat shock protein 90 (HSP90) (Figure 2A, right). Collectively, these additional stains revealed no significant differences between control and 2A-treated lysates, suggesting that ACTB may be a specific target.” (Page 7)

The authors briefly mention that it is odd that one of the proteolytic targets is ATG4A, even though autophagy is widely considered a proviral process for these types of viruses. They do explain that this may have more to do with timing, but I feel this explanation is somewhat weak. Further elucidating why turning off autophagy late in infection could be beneficial would enhance the narrative overall. It has been reported that autophagy/flux is important for viral maturation early on, but later in the viral life cycle, autophagic flux seems to be detrimental for the virus (ie-extracellular vesicle release is impaired). Perhaps there might be something there that could explain this phenomenon. Otherwise, again, 2A is very promiscuous, so ATG4A could just be collateral damage.

Response: We thank the reviewer for these comments which we have clarified through additional discussion in the revised manuscript. “Although it is possible that ATG4A may be a collateral target of the promiscuous 2A protease, findings from the current study clearly demonstrate an impaired functional autophagy through abnormalities in LC3 lipidation upon the expression of cleaved T-ATG4A supporting a potential strategy by which virus may disrupt canonical autophagy functions through direct targeting of autophagy factors” (Page 13).

Reviewer 2 Report

This manuscript is a well written account of some straightforward elegant experiments. The authors have convincingly demonstrated that the host autophagy related protein ATG4A is specifically targeted for proteolytic degradation during Coxsackie B3 virus infection. Moreover the viral protease involved has been identified by in vitro cleavage analyses and the site of cleavage identified by sequence related prediction and confirmed by demonstration that a altered version of the protein bearing a mutation at the cleavage site was, in fact resistant to proteolysis. These virological/biochemical aspects of the study are clear and the results unequivocal. However the virological/cell biological aspects are more complicated to understand as ATG4A and its cleavage seem to function differently at different stages in the viral replication cycle. I appreciate that the study has opened a complex 'can of worms' at this stage which will need further work to unravel. However, there are a couple of points about the study that I would liked to see addressed/commented upon by the authors:-

1) A flag tagged version of ATG4A (and other proteins) was used for obvious pragmatic reason of easy detection. It is unlikely (but not impossible) that the tagged protein has slightly different properties to the native form and this should be at least commented upon.

2) The cleavage site mutant ATG4A has clearly resistant to 2A cleavage in vitro. Would it make sense to investigate the consequences of transfecting cells with this variant protein in parallel with the native form?

Author Response

Reviewer #2

This manuscript is a well written account of some straightforward elegant experiments. The authors have convincingly demonstrated that the host autophagy related protein ATG4A is specifically targeted for proteolytic degradation during Coxsackie B3 virus infection. Moreover the viral protease involved has been identified by in vitro cleavage analyses and the site of cleavage identified by sequence related prediction and confirmed by demonstration that an altered version of the protein bearing a mutation at the cleavage site was, in fact resistant to proteolysis. These virological/biochemical aspects of the study are clear and the results unequivocal.

Response: We thank the reviewer for these supportive comments.

However the virological/cell biological aspects are more complicated to understand as ATG4A and its cleavage seem to function differently at different stages in the viral replication cycle. I appreciate that the study has opened a complex 'can of worms' at this stage which will need further work to unravel. However, there are a couple of points about the study that I would like to see addressed/commented upon by the authors:-

1) A flag tagged version of ATG4A (and other proteins) was used for obvious pragmatic reason of easy detection. It is unlikely (but not impossible) that the tagged protein has slightly different properties to the native form and this should be at least commented upon.

Response: We thank the reviewer for this insightful comment regarding the potential effects of Flag-tag on protein function which we have addressed in the revised manuscript. “In the current study, we utilized a Flag-tagged ATG4A, and we cannot exclude the possibilities that 3 Flag tag has other impact. However, in our previous research, the use of a 3 Flag tag did not interfere with protein function likely due to the small molecular size (~ 3kDa) of the tag” (Page 14).

2) The cleavage site mutant ATG4A has clearly resistant to 2A cleavage in vitro. Would it make sense to investigate the consequences of transfecting cells with this variant protein in parallel with the native form?
Response: We thank the reviewer for this suggestion which we have addressed experimentally in the revised manuscript (Figure 4C). Briefly, the transfection of a non-cleavable mutant (G374E) or truncated ATG4A, did not significantly impact viral protein production as compared to wildtype ATG4A expression although all three constructs showed a downtrend compared to vector only control.

Reviewer 3 Report

In the manuscript entitled “Coxsackievirus Protease 2A Targets Host Protease ATG4A to Impair Autophagy”, the authors depict the pro-viral function of ATG4A through genetic silencing, overexpression and endogenous protein expression. This study could contribute towards understanding how enteroviruses destabilize the autophagy pathway. 

Here are few suggestions for the authors to improve the manuscript.

Line 13, Enterovirus should be replaced by Enteroviruses

Line 22, cleavages should be replaced by cleaves

Line 26, shed should be replaced by sheds

Line 56, starting “This complex”, references are missing

Lines 84-87 are repetitions from lines 81-84

Lines 93-94 are repetitions from lines 91-92

Line 185, 1’ SDS should be replaced with 1% SDS

Line 128, contains editing errors 5.5´105

Figure 1A. The authors report decrease in ATG4A protein expression following CVB3 infection. Can the authors show that there were no changes in the gene expression via qPCR?

Figure 2A. Can the authors explain the difference between actin expression?

Figure 3A. Does ATG4A knockdown have any effects on cell growth? Would there be an effect on viral titers? 

Figure 3E. This figure is not well explained in the text. In figure 3B, the authors use viral 2A to depict effect on viral RNA, whereas in figure 3E, the authors use viral VP1 gene, is there a reason for that?

Also, its not clear from Figure 3E, does the effect of wt-ATG4A and T-ATG4A overexpression similar to vector only transfected cells?

As shown in Figure 3C, does T-ATG4A overexpression have any effect on VP1 protein expression?

Figure 4B. The input shows a very weak expression of ATG4A and T-ATG4A. The IP lane shows very faint expression of T-ATG4A, so its hard to say that GABARAP shows weaker binding with T-ATG4A.

Also, it will be interesting to see the effect of a deletion in any other region of ATG4A as a control.

Author Response

Reviewer #3

In the manuscript entitled “Coxsackievirus Protease 2A Targets Host Protease ATG4A to Impair Autophagy”, the authors depict the pro-viral function of ATG4A through genetic silencing, overexpression and endogenous protein expression. This study could contribute towards understanding how enteroviruses destabilize the autophagy pathway. 

Here are few suggestions for the authors to improve the manuscript.

Line 13, Enterovirus should be replaced by Enteroviruses

Response: This has been corrected.

Line 22, cleavages should be replaced by cleaves

Response: This has been corrected.

Line 26, shed should be replaced by sheds

Response: This has been corrected.

Line 56, starting “This complex”, references are missing

Response: We thank the reviewer. The necessary references are now included.

Lines 84-87 are repetitions from lines 81-84

Response: We thank the reviewer for catching this error which has now been corrected.

Lines 93-94 are repetitions from lines 91-92

Response: We thank the reviewer for catching this error, this has now been corrected.

Line 185, 1’ SDS should be replaced with 1% SDS

Response: This has been corrected.

Line 128, contains editing errors 5.5´105

Response: This has been corrected.

Figure 1A. The authors report decrease in ATG4A protein expression following CVB3 infection. Can the authors show that there were no changes in the gene expression via qPCR?

Response: We thank the reviewer for suggesting this important control which is now included in the revised manuscript. “Furthermore, we tested the gene expression levels of ATG4A in sham and CVB3-infected cells and confirmed that gene expression was unaltered between sham and CVB3 at a timepoint of infection (7hrs pi) when ATG4A protein was declining (Figure 1C). “   

Figure 2A. Can the authors explain the difference between actin expressions?

Response: We thank the reviewer for this question which we have clarified in the revised manuscript with the support of additional experimentation. “Given the significant alteration in ACTB protein expression following 2A treatment, we further assessed the total protein levels following in vitro cleavage using Ponceau S stain as well as an additional housekeeping protein, heat shock protein 90 (HSP90) (Figure 2A, right). Collectively, these additional stains revealed no significant differences between control and 2A treated lysates, suggesting that ACTB may be a specific target.”

Figure 3A. Does ATG4A knockdown have any effects on cell growth? Would there be an effect on viral titers?

Response: We thank the reviewer for this comment which we have addressed with additional experimentation. “As a control, we assessed the viability of cells following 48h silencing of ATG4A and observed no significant cell death as compared to si-control treated cells (Figure 3C).” and “ATG4A-silenced cells also showed decreased quantity of viral RNA (as measured with viral 2A primers) and viral titers compared to cells treated with control siRNA, suggesting a pro-viral function for ATG4A (Figure 3B).”

Figure 3E. This figure is not well explained in the text. In figure 3B, the authors use viral 2A to depict effect on viral RNA, whereas in figure 3E, the authors use viral VP1 gene, is there a reason for that?

Response: We thank the reviewer for this comment which was addressed in the revised manuscript. “We used both viral genes 2A (Figure 3B) and VP1 (Figure 3F) to measure viral RNA because CVB3 utilizes a single open reading frame encoding a single-copy of each viral gene. This unique characteristic of monopartite viruses such as CVB3 allows for the accurate quantitation of viral RNA irrespective of which viral gene is used for RT-qPCR.”

Also, its not clear from Figure 3E, does the effect of wt-ATG4A and T-ATG4A overexpression similar to vector only transfected cells?

Response: Thank you, this has been clarified in the revised manuscript. “… quantitative assessment of viral RNA using viral VP1 primers demonstrated no significant changes between control, WT-ATG4A, and T-ATG4A expressing cells, suggesting that both WT-ATG4A and T-ATG4A do not significantly perturb viral RNA production (Figure 3F)”.

As shown in Figure 3C, does T-ATG4A overexpression have any effect on VP1 protein expression?

Figure 4B. The input shows a very weak expression of ATG4A and T-ATG4A. The IP lane shows very faint expression of T-ATG4A, so it’s hard to say that GABARAP shows weaker binding with T-ATG4A.

.

Response: We thank the reviewer for this comment which we have addressed in the revised manuscript through additional experimentation and discussion (Page 14). “Finally, we sought to test whether ATG4A mutants had direct impact on viral protein production. To that end, HeLa cells were transiently transfected with constructs encoding either wildtype, truncated, or non-cleavable mutant ATG4A. Interestingly, no significant alterations in viral protein production was observed among these three constructs (Figure 4C)“ (Page 11).  Additionally, the revised manuscript now includes stronger IP results for T-ATG4A (Figure 4B).

Also, it will be interesting to see the effect of a deletion in any other region of ATG4A as a control
Response: We thank the reviewer for this suggestion. LIR region of ATG4A was deleted in this study because the resulting truncated fragment mimicked the cleaved fragment that was observed following CVB3 infection and confirmed by mutagenesis studies. Since we identified the cleavage site and cloned the corresponding cleavage-resistant mutant at these particular loci, we chose to elucidate and characterize these mutants in the current study. 

Round 2

Reviewer 1 Report

The authors sufficiently addressed previous concerns.